# Economic Trading Susceptibility: Constructing Networks of Mutual Influence through the Fitness of Countries

**DOI:** 10.3390/e25010141

**Published:** 2023-01-10

**Authors:** Nishanth Kumar, Henrik Jeldtoft Jensen, Eduardo Viegas

**Affiliations:** 1Centre for Complexity Science and Department of Mathematics, Imperial College London, South Kensington Campus, London SW7 2AZ, UK; 2Institute of Innovative Research, Tokyo Institute of Technology, 4259, Nagatsuta-cho, Yokohama 226-8502, Japan

**Keywords:** evolutionary dynamics, graph and networks, economy

## Abstract

The emergence of economic blocks and the level of influence countries exert on each other are fundamental features of the 21st century globally interconnected economy. However, limited quantitative research exists measuring the level of influence among countries and quantitatively determining economic blocks. This research develops a method to quantify the mutual influence of countries by making use of relatively standard procedures for complex networks in order to assemble non-trivial networks of influences and to identify symbiotic relationships. The methods are of significant help to an enhanced understanding of the global politics of trading and associations. Moreover, we develop the Mutual Influence Robustness (MIR) metric to work together with the Economic Fitness metric to provide some level of predictive modeling for the trends and future paths of countries. Our key results show the existence of a mutually influencing network around East and Southeast Asia, developed North America, and the northern and Iberian countries. Moreover, we find that it is possible to do some level of path predictability for the fitness and mutual influence of countries.

## 1. Introduction

Within economics, there is a high level of debate as to whether a given country ought to obtain a competitive advantage by a strategy centred on product specialisation or whether it should aim to achieve a portfolio approach to products. The Riccardian approach to measuring the competitiveness of economies suggests that the wealthiest countries should produce only a few products with a high degree of specialisation that does not overlap with each other provided that a level of free trading agreements exist among these economies [1,2].

A contrasting, but not necessarily conflicting, approach adopted by others is to categorise products into different ‘added value’ bands and to argue that specialisation is happening towards the ‘high end’ grouping, but that a diversity of products within that range of products is adopted [3,4,5,6].

The notion of product specialisation was later challenged by the works of Hidalgo and Hausmann (HH) [7] through the introduction of the ‘Method of Reflections’ analysis. Their method was inspired by concepts traditionally associated with Graph and Network Theory, becoming the cornerstone for some research that later became associated with the broader concept of ‘Economic Complexity’ [8,9,10,11]. The method essentially (a) interpreted the existing data for the country product space as bipartite networks in which countries are connected to the products they export, (b) derived an adjacency country–product matrix representing a network of interactions, and (c) interactively calculated a symmetric set of variables for the two types of nodes in the network (countries and products) which can represent the observed levels of diversification of a country as well as the ubiquity of a product. Ultimately, their result indicates that competitive countries tend to produce all products that are possible, given their level of technology, and not just only technologically “sophisticated” products.

The method of reflection used by Hidalgo and Hausmann was adapted to take into account more specific measures of complexity to quantify the “competitiveness” of an economy. Central to this research is the argument that the diversification of production capabilities can increase competitiveness. The work within this area challenges the canonical economic narrative, where the economic competitiveness of countries is measured in monetary metrics such as the Gross Domestic Product (GDP). Specifically, their approach creates two measures: the observed levels of diversification of a country and the ubiquity of a product. The results have been used, in tandem, to rank countries based on their export profile [7]. Importantly, the Economic Complexity measures introduced by [7] incorporate the concept of competitiveness of a country underpinned by the construction of a linear relationship between the complexity of the product and the average fitness of the countries that produce it. This has raised issues as to whether the results from this approach truly capture the concept of competitiveness [9,12].

In order to address some of the perceived weaknesses of the Economic Complexity framework’s ‘linear’ approach, an enhanced method for the measurement of economic complexity was developed by Pietronero, Tacchella, Calderalli and their collaborators [9,10,11,13,14,15]. This research [13] makes use of crude oil as an anecdotal example of the inconsistency of Economic Complexity as a measure to rank the competitiveness of countries. It argues that the production of crude oil is dependent almost entirely on the availability of the resource, and therefore, countries cannot ‘free willingly’ dedicate financial resources to start producing the product. However, by mathematical construct, the method of reflection tends to result in attaching a high-ranking status to countries that are significantly dependent on crude oil as the primary export. Ultimately, such an approach leads to a counter-intuitive, and arguably economically unrealistic, fitness ranking order where Qatar is placed well above China.

The framework is similar to that of HH’s to the extent that it makes use of a bipartite network to obtain a set of measures for two types of nodes (countries and products). As a result, the “fitness of a country” and the “complexity of a product” are calculated and measured (as opposed to HH’s original metrics of “diversity of a country” and the “ubiquity of a product”). However, it substantially differs in the method as (a) it makes use of a distinct set of weighting variables, and (b) the original symmetry of the formulae constructs between the two measures is broken. Essentially, this means that a country’s fitness is weighted by the complexity of its exported products, whereas the complexity of a product itself is inversely proportional to the number of countries exporting the selected product. The method leads to products produced by low-fitness countries having a lower level of complexity due to the level of technology needed by them [12,13,16].

In addition, the marginal change in the fitness of a country that produces multiple products would be higher as the technological innovations required would be higher. As a result, it can be argued that high-fitness countries give limited information on the complexity of a product because these countries export almost all products.

Whereas we believe that research and work on the best methods to measure a country’s competitiveness and/or fitness are highly valuable to the promotion of academic knowledge, our work has a substantially distinct motivation: we are particularly interested in (a) measuring the mutual influence among countries, (b) quantifying the susceptibility for these countries as a result of the derived network of influences, and (c) looking at the evolution of fitness and susceptibility from a time series, evolutionary trending, perspective.

Here, we believe that it is important to dwell on some level of detailed semantics with regard to our usage of the term ‘mutual influence’ in order to avoid confusion and misunderstanding. In social sciences, influence is normally associated with the concept of power to make people agree with one’s opinion. However, we are not dealing with the concept of power in isolation. Instead, we are simply using the term as a way to synthesise a mathematical construct: ‘mutual influence’ captures the existing symmetry in the absolute change to the fitness of a country with the removal of another country within our population (and vice versa).

In a similar manner, the term ‘fitness’ is solely related to the quantitative values obtained from the economic fitness method [10], and it should not be taken as any other more generic definition. Therefore, in this paper, we simply opted to preserve, replicate and make use of such a method to quantify the fitness of countries. The novel element of our research consists therefore in deriving additional methods and calculations based on the changes to the metric instead of the metric itself. We judge the original framework to be well suited as the basis for our approach and method given that fitness is a relative quantity that is not measured in isolation, and it is dependent on the indirect level of interaction with the other agents within the system. Moreover, the ‘game’ or interaction rules are defined by simple mathematical equations without extensive use of parameters. Having said the above, we will purposely stay away from elements that we judge to be less attractive within the overall field of Economic Complexity. In particular, we focus on single measurements (i.e., the fitness in this case) as predictors for noisy economic measures such as the Gross Domestic Product (GDP).

Having stated our perspective and usage of the terms ‘mutual influence’ and ‘fitness’, we summarise the key methodological foundation of our research: the relative influence of a country on all others can be essentially captured by its removal from the data and by observing the resulting changes in the fitness of all other remaining countries. Expanding on this concept, we can mathematically observe that the inherent recursive property of the economic fitness method [10] leads to non-linear and non-trivial changes to both the complexity of all products as well the fitness of all other countries if a single country is removed from the data. As a result, the normalised changes to the ‘fitness’ of a country can be taken as a theoretical proxy for the effect, or ‘influence’, a country has onto another. We emphasise here, however, that there are two aspects within the real economic world that are not captured by the method. Firstly, there is the obvious fact that a country simply cannot disappear (i.e., full removal). Instead, it is more likely that it may stop producing and/or exporting certain items. Secondly, as typical as in any complex system, a ‘rewiring’ would be likely. In other words, if a country were to disappear, another one could feasibly step in and start producing/exporting to compensate for the loss of that country. We opted here to ignore these elements in order to ensure that we focus on the essential dynamics of influence from a historical data perspective, since we do not aim to project any future evolutionary changes to the global economic structure.

Lastly, we make note that our framework is inspired by the Tangled Nature Model [17] where an agent fitness is effectively dependent on the overall population of species within the given ecological and economical space of interactions [18].

Moreover, with regard to methods, we attempted to make use of standard procedures for complex networks by gradually removing edges and nodes in different ways and orders to quantify sensitivities and susceptibilities. We would also regard the assembling of the influence network as a standard practice inspired by adjacent matrix methods [19,20].

## 2. Materials and Methods

### 2.1. Original Fitness Algorithm and Data

As previously described, we make use of the Economic Fitness and Complexity method, which is described in detail in [10] as the basis to calculate the fitness of countries, $F_{c}^{\left(n\right)}$.

The calculation is based on a recursive process through an algorithm consisting of two core formulae that for the nth interaction gives the fitness $F_{c}^{\left(n\right)}$ of country *c* as well as the complexity $Q_{p}^{\left(n\right)}$ of product *p*: (1)$$\begin{array}{cc} & \left\{\begin{array}{c} {\bar{F}}_{c}^{\left(n\right)}=\sum_{p\in P}M_{cp}Q_{p}^{\left(n-1\right)}\\ {\bar{Q}}_{p}^{\left(n\right)}=\frac{1}{\sum_{c\in C}M_{cp}\frac{1}{F_{c}^{\left(n-1\right)}}} \end{array}\right. \end{array}\begin{array}{c} \left\{\begin{array}{c} F_{c}^{\left(n\right)}=\frac{{\bar{F}}_{c}^{\left(n\right)}}{<{\bar{F}}_{c}^{\left(n\right)}{>}_{c}}\\ Q_{p}^{\left(n\right)}=\frac{{\bar{Q}}_{p}^{\left(n\right)}}{<{\bar{Q}}_{p}^{\left(n\right)}{>}_{p}} \end{array}\right. \end{array}$$

From the equations in Equation (Equation 1), it can be noted that the iterative algorithm is composed of two steps: computing $\bar{F}$ and $\bar{Q}$ at every iteration and then normalising $\bar{F}$ and $\bar{Q}$ to compute the final fitness *F* and complexity *Q* vectors. The primary input in Equation (Equation 1) is the matrix $M_{cp}$, which represents the country–product mix that holds a value for the competitiveness of a country *c* in producing product *p*. More specifically, two different versions for $M_{cp}$ were used in previous research: (i) the extensive fitness, where $M_{cp}$ reflects the market shares of exported products [14], and (ii) the intensive fitness, where $M_{cp}$ is made up of binary entries 0 or 1 depending on whether a country is deemed to have a relative comparative advantage in the production of a product [7,21]. In our work, we opted to use the extensive fitness approach as we judged it to be more consistent with typical measures of diversity where the existence, as well as the abundance of species, are important factors.

Data to source the matrix $M_{cp}$ is obtained from the Integrated Database of Economic Complexity (IDEC) [14]. The database provides an integrated source of information on the production of goods and services for 160 countries and a total of 124 sectors, from 1996 to 2018.

### 2.2. Assembling the Countries’ Influence Matrix

By adopting the fitness algorithm in Equation (Equation 1) without any modifications, we are able to assemble a matrix that represents the influence, or the effect, that one country $c^{*}$ has on another country *c* through the following steps:

*Step* I: The fitness, $\Phi_{c}$, of a country *c* is calculated in accordance with Equation (Equation 1). Here, $\Phi_{c}$ is equivalent to $F_{c}$. We only make a distinction to emphasise that the former is calculated by us, whereas the latter refers to the original equations and method.

*Step* II: A different country, $c^{*}$, is removed; i.e., the exports of this country are made to be 0 and the market share of every product is recalculated.

*Step* III: The new fitness $\Phi_{c\backslash c^{*}}$, without country $c^{*}$, in relation to the original country *c* is calculated.

*Step* IV: The initial fitness of the country, $\Phi_{c}$, is renormalised so that: (2)$$\hat{\Phi_{c}}=\frac{\Phi_{c}\sum_{1}^{N}\Phi_{i}}{\sum_{1}^{N}\Phi_{i}-\Phi_{c\backslash c^{*}}}$$

*Step* V: The influence of a country $c^{*}$ on country *c* is then computed as: (3)$$\delta \phi_{c^{*}\to c}=\left|\hat{\Phi_{c}}-\Phi_{c\backslash c^{*}}\right|$$

*Step* VI: The process is repeated for the combination of all countries *c* and $c^{*}$, leading to the formation of a dense matrix $I_{c^{*}c}$.

It is worth noting that from a semantic perspective, we decided to describe $\delta \phi_{c^{*}\to c}$ as “influence” instead of “dependency” in order to avoid undue association with reliance or control that are typical of the latter. Instead, the computation in question aims to simply capture the effect of one country on to the fitness of another, which is an aspect in closer alignment with the former. This is also the primary reason for considering the absolute value of δϕ and ignoring the sign.

From a methodological perspective, it is also important to point out that the mathematical form in Equation (Equation 3) is underpinned by the principle of a zero-sum game, where the non-absolute $\sum_{i=1}^{N}\sum_{k=1}^{N}\left(\hat{\Phi_{i}}-\Phi_{i\backslash k^{*}}\right)=0$. Therefore, the total ‘fitness’ value is preserved and the ‘influence’ is essentially a reallocation of these values. As a result, the functional intends to capture the shuffling of the fitness as opposed to the relative changes to the fitness.

### 2.3. Constructing Networks of Mutual Influence

The Countries’ Influence Matrix $I_{c^{*}c}$ can be essentially described as a fully connected network (i.e., all nodes are connected to each other). In order to construct Networks of Mutual Influence $I_{{c^{*}c}_{\left(\tau \right)}}$ at distinct levels, we adopted the classical complexity and network method of removing edges in an ordered, non-uniform, manner [22] by varying the threshold level, τ, for which an edge is deemed to exist. The matrix representing a specific Network of Mutual Influence for a given τ is obtained as follows:(4)$${I_{c^{*}c}}_{\left(\tau \right)}=\begin{array}{c} \left\{\begin{array}{c} 1\;\Leftrightarrow \;\delta \phi_{c^{*}\to c}>\tau \;and\;\delta \phi_{c\to c^{*}}>\tau \\ 0\;otherwise \end{array}\right. \end{array}$$

The removal rules from Equation (Equation 4) essentially mean that as τ increases, the symbiotic relationships of commensalism (i.e., one country’s influence is neutral, or small, within the pairing) tend to be removed at early stages. Effectively, as τ increases, the preserved edges will tend to reflect symmetrical relationships of mutualism, competition, parasitism, or reciprocal altruism (i.e., those edges where $\phi_{c^{*}\to c}$ and $\phi_{c\to c^{*}}$ have both higher and similar values).

### 2.4. Mutual Influence Robustness (MIR) Metric

The MIR metric for each country is obtained by: (a) gradually increasing τ to generate a series of $I_{{c^{*}c}_{\left(\tau \right)}}$ matrices in accordance with Equation (Equation 4) and counting the number of existing edges, $k_{c}$, connected to each country in each τ realisation (i.e., the degree $k_{c,\tau }$ at level τ); (b) plotting a graph of $k_{c}=f\left(\tau \right)$; and (c) calculating the metric, represented by $\chi_{c}$, corresponding to the area under the curve:(5)$$\chi_{c}≃\sum_{i=0}^{N}k_{c,\tau_{i}}\Delta \tau_{i}$$
where [0,T] is partitioned into *N* parts such that $\tau_{0}=0$ and $\tau_{N}=T$, and $\Delta \tau_{i}=\tau_{i}-\tau_{i-1}$.

Intuitively, smaller values of χ tend to reflect the countries where the influence tends to be weighted towards commensalistic relationships, where the level of engagement is much less reciprocal. In contrast, countries with higher values of MRI will tend to be highly interweaving.

### 2.5. Generating the Group of 20 Mutually Influenced Countries (M_*20*_)

The central cluster of $M_{20}$ is obtained by the same removal rules from Equation (Equation 4) through the gradual increase of τ with the distinction that it stops when the network is fragmented and the largest cluster contains 20 nodes exactly.

## 3. Results

Our results are divided into three separate but complementary sections. Firstly, we summarise our core findings in relation to the relationship between the countries’ fitness and their associated MIR, namely the mutual influence robustness. We follow on by providing a perspective on the historical paths and trends of these two metrics. In the Section 3.2, we move to a more detailed analysis of individual countries and provide a geographical perspective of MIR and fitness from a comparative ranking approach. Lastly, we carry out a comparative analysis between the $M_{20}$ (i.e., the dominant group of mutually influenced countries), the group of the 20 countries with the highest fitness, the $\Phi_{20}$ and the traditional economic blocks known as the G7 and G20.

### 3.1. MIR and Fitness: Observations and Trends

A scaling relationship can be observed from Figure 1 where $\chi \propto \phi^{1/2}$. Such a relationship is maintained over the 18-year period of data between 2000 and 2018. Although these relationships are neither exact nor trivial, we find that it is the perpendicular distance δ from the fitting line that provides very valuable insight into the likely future trends of fitness and MIR development paths. It can be observed from Figure 2a.1,a.2,d that countries that have a distance δ above a threshold of 0.2 from the average fitting line tend to increase their distance δ even further.

Moreover, as it can be noted in Figure 2d, the dispersion for the direction as well as the average yearly distance tends to be narrow and therefore relatively predictable. In contrast, countries with distance δ below a threshold of 0.2, Figure 2c.1,c.2,f, have a very large dispersion and more erratic paths both in terms of direction and distances of movements. Therefore, it is much harder to predict both the individual as well as the average paths of these countries. Lastly, countries with smaller δ distances, Figure 2b.1,b.2,e, move equally in both directions, above and below the fitting line, albeit with a small average tendency for increasing of the distance δ over time.

### 3.2. A Geographical Perspective of MIR and Fitness

The rankings of countries based on different metrics such as fitness, economic complexity index, GDP, per capita GDP and others have always been a contentious debate within both academic and general economic circles. However, it is not our intention with this research to provide another set of ranking results. Instead, the results that we focus on here are the fact that regional geographical clustering can be clearly observed by analysing the gap between the fitness and MIR rating. We select three regions here for specific comments.

Firstly, Eastern European, and to a smaller extent Mediterranean, countries tend to have a fitness ranking much higher than the corresponding ranking based on MIR. These results suggest that although these countries have a relatively sophisticated level of product complexity, their influence within the broader global trading network is somehow limited. Here, we would tentatively speculate that this situation may well be a consequence of these countries being effectively satellites of the European common market, where their products and exports are shaped to reciprocally fit the needs of the core, larger, European economies.

In contrast, we note that African countries in general, as well as smaller South American countries, tend to have a fitness ranking at much smaller levels than their MIR, with the examples of the Democratic Republic of Congo and Bolivia as the most extreme ones. Fitness and economic complexity rankings for these countries tend to be very low given that their exports are mainly based on natural resources. However, the MIR gives another perspective on the trading profile of these countries: These natural resources tend to be produced by other similar economic-leveled countries, who are often neighbors (i.e., geographical correlation) but do not tend to be produced (or at least exported) by the larger countries who are normally customers. As a result, their mutual influence tends to be higher due to their interplay at economically similar, and/or regional, levels.

The third region to comment on is East Asia, where most of the economies also tend to have a pronounced higher level of MIR ranking when compared to their fitness. This is in particular the case for Vietnam, Cambodia, and Bangladesh. Here, the MIR unveils a perspective: These countries have a proportionally higher level of mutual influence with larger economies such as China, Great Britain, the USA, and India.

Lastly, whereas it is not our intention to fully explore the relationship between MIR and GDP as typically performed within economic complexity research [10,11,14,15] we carried out a limited evaluation as to whether the ranking gap analysis for the countries cited above is broadly consistent with the GDP growth performance of these countries for the period. The GDP data are sourced from the World Bank website, and the compound growth was calculated based on the GDP (constant 2015 US$). Starting with Vietnam, Cambodia, and Bangladesh, these countries grew at an average of 6.5%, 6%, and 7.6% a year over the 2000 to 2018 period, which is well above the East Asia and Pacific average of 5.1%. In contrast, as it can be observed in Figure 3, Thailand, which has a contrary position where the fitness ranking is significantly above its MIR ranking, had an average GDP yearly growth of solely 4%.

Within Sub-Saharan Africa, the Democratic Republic of Congo and its neighbor, Zambia (both highlighted in darker blue), in Figure 3 grew at an average of 5.4% and 6.2% a year, which is well above the 4.4% average for the region. In contrast, the darker red, South Africa, had a relatively meager average growth of 2.7%. Lastly, within South America, the Andean and landlocked countries (most of the blues) experienced an average growth between 3.8% and 5% a year and therefore at higher levels than their neighbors, Brazil, Argentina, and Colombia that experienced growth of 2.3%, 2.2% and 3.8%, respectively.

These results combined suggest a better relationship between the fitness ranking and the country’s GDP might be achieved if MIR were to be used as a complementary metric to fitness.

### 3.3. The Group of 20s Mutually Influenced Countries

By assembling the group of 20s mutually influenced countries $M_{20}$, we are able to extract some insights in relation to the potential importance of countries and the nature of their symbiotic relationships.

Firstly, panel (a) within Figure 4 shows a highly connected central cluster consisting of the G5 founding members (i.e., USA, United Kingdom, France, Germany, and Japan) plus China, which is defined here as the $M_{5+1}$. All nodes are connected to each other with the sole exception of France and Japan. This shows that our assembled $M_{20}$ network results in the $G_{5+1}$ to be the central cluster of global mutual influence. In addition, by observing panel (c), it can be noted that all relationships within the G5 group are of our definition of a mutualistic nature, i.e., the fitness of both countries reduces without the existence of the other. However, the symbiosis with China is of a very different nature: either it is a competition, i.e., the fitness of both countries is higher without the existence of the other, or it is a reciprocal altruism such as in the case of Japan and Germany where their fitness is reduced without China, but China’s fitness is better without either. It is tempting to speculate that the relationship between these major exporting economies is distinct due to the fact that Japan and Germany might well depend on China for basic components, whereas China has a fully integrated supply chain. Most importantly, our results show the distinct nature of the relationship between the G5 and China, which reflects the politics in the real world where the interests between the G5 tend to be aligned but at a number of times at odds with those of China.

In addition, by comparing group memberships, it is possible to note that there is a significant overlap between the $M_{20}$ and the countries with the highest fitness. However, there is a marked difference to the real-world group of the G20. Whereas the $M_{20}$ results show a higher emphasis on the East and Southeast Asian countries and European economies, the G20 contains countries of other continents and regions (such as Argentina, Brazil, Mexico, South Africa, Australia, Saudi Arabia, and Russia). Our results suggest that whereas these economies may play a significant role within their regions, they are not mutually influential on a global scale. In conclusion, mutually influential economies are centred within the northern hemisphere and clustered in (a) East and Southeast Asia; (b) northern and Iberian Europe; and (c) developed North America.

When analysing the changes for the $M_{20}$ network over the 18-year period of data, as shown in Figure 5, one can notice three revealing aspects. Firstly, the changes to the membership are fairly limited, with a likely tendency to add East Asian countries to replace the European countries, as shown in panels (a) and (d). Secondly, the middle panels (b) and (e) also indicate the loss of mutual influence of the US that is almost exclusively reduced to the core $G_{5+1}$. In contrast, one can observe China continuing to be, if not expanding, the main countries connecting to the outer core beyond the $G_{5+1}$. Lastly, the right panels (c) and (f) show the evolution of the $G_{5+1}$ from a relatively sparsely connected group to a highly connected position in 2018.

## 4. Discussion

In this research, we have shown that by switching focus from specific computation methods for fitness and economic complexity indices to the changes to their relative measures, it is possible to construct networks of influence that are consistent with real-world observations and that unveil specific phenomena and dynamics that were previously qualitatively described but not quantitatively observed. For example, our work provides a clear framework that shows the relative retraction of the USA from global partnerships over the last 20 years, the nature of the cooperation of the G5 and their competition with China, and the continuing emergence of East and Southeast Asia as an area of significant economic influence. By making use of relatively standard procedures for complex networks to quantify sensitivities and susceptibilities, it is possible to assemble a non-trivial network of influences and identify symbiotic relationships that are of significant help to an enhanced understanding of the global politics of trading and associations. Moreover, our results suggest that a better understanding of a country’s strength and profile can be achieved by the computation and implementation of the Mutual Influence Robustness (MIR) metric to work together with the Economic Fitness metric. In particular, the joint analysis of these metrics may help with some level of predictive modeling.

Having said the above, we highlight here, however, that additional insight may be obtained if one were to take into account the effect of the fitness of a country not only based exclusively on its exports profile but also on its imports profile. We would speculate that further work might potentially unveil a better appreciation of the underlying dynamics of trading and economic performance (a) by combining the Fitness and MIR metrics, through associations at global and local level network interactions, and (b) by incorporating the import profiles in tandem with the existing export profiles. We would, therefore, regard this as an area for future work.

## Figures and Tables

**Figure 1 entropy-25-00141-f001:**
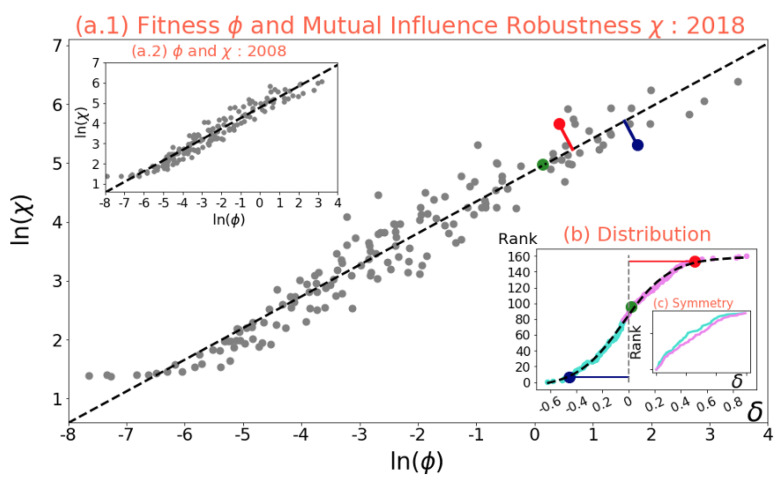
Relationship between the Countries’ Fitness and the Mutual Influence Robustness. Plot (**a.1**) and inset (**a.2**), both drawn on a natural logarithm scale, show the scaling relationship between the fitness of the countries and their associated MIR, for 2018 and 2008, respectively. Each gray dot represents a country, whereas the dashed black lines consist of the linear best fitting for all countries within the year, with an exponent α≈1/2. The colored dots—red, green, and blue—are illustrative examples of countries well above the fitting line, at the line, and well below the line respectively, whereas the corresponding colored lines represent the perpendicular distance from the fitting line δ. The countries selected for illustration are Vietnam, Australia, and Italy, respectively. The inset (**b**) shows the distribution of the perpendicular distance from the fitting line δ within the x-axis, ranked from the largest below the line to the largest above the line, the y-axis. The dotted black line is the best fitting of the data by making use of a standard logistic curve. The addition inset (**c**) is a visual representation of the existing symmetry for the absolute values of the distances from the fitting line for the ranking of countries below the line (cyan) and that of those above the line (pink).

**Figure 2 entropy-25-00141-f002:**
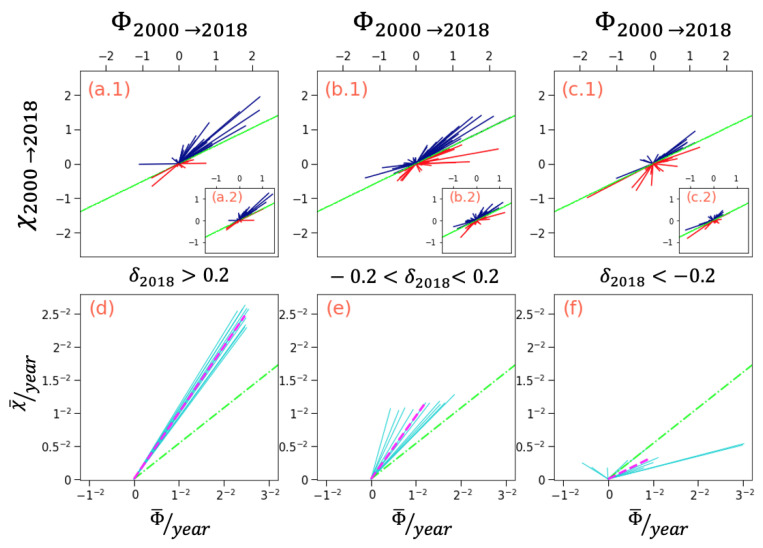
Individual paths and average paths of countries within the fitness vs. MIR axes. Plots (**a.1**–**c.2**) show the straight path of countries, each one represented by a single line, from the year 2000 (2008 for the insets) to 2018. All paths are set at the (0,0) point of origin. The green lines for all plots represent the χ=αΦ baseline where α≈1/2 corresponds to the slope derived from Figure 1. Blue and red lines represent countries that trended above and below the baseline, respectively. The plots represent three distinctive cohorts: Countries with a perpendicular distance δ from the fitting line above 0.2 (**a**–**d**), countries with a distance δ below 0.2 (**c**–**f**), and those between (**b**–**e**). Plots (**d**–**f**) represent the average yearly dislocation of all countries within the cohort. The cyan lines represent data from different eleven distinctive time windows ranging data from (2000 *…* 2010) to 2018. The dotted purple line represents the average of all lines for the relevant cohort.

**Figure 3 entropy-25-00141-f003:**
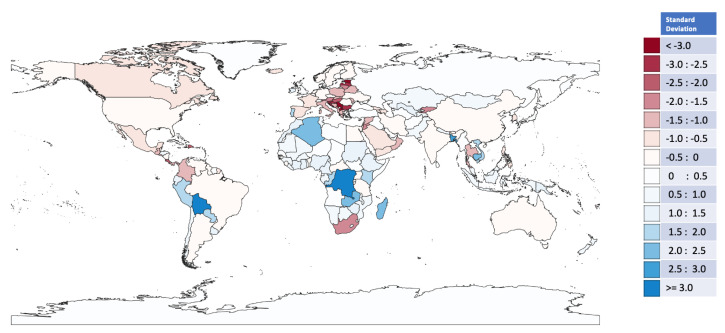
World map of the gap between the rankings of countries’ fitness and MIR. The color—coded map shows the distribution of the countries based on the highest differences between the fitness ranking derived from Equation (Equation 1) and the MIR ranking derived from Equation (Equation 5). The colors reflect the mismatch between the Fitness and MIR rankings. The most aligned rankings are those nearest to white; the darker the color, the higher the misalignment. Shades of blue represent countries where the MIR ranking is above fitness, whereas shades of red represent the reverse. The standard deviation was automatically calculated by the QGIS software, version 3.6.

**Figure 4 entropy-25-00141-f004:**
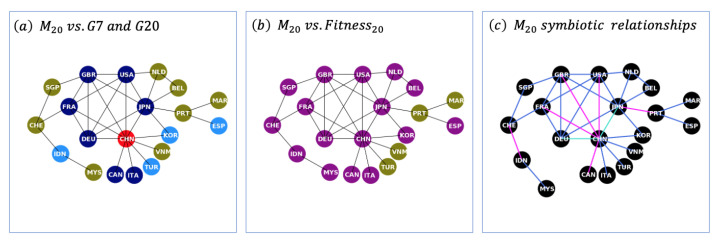
Comparative Analysis: The Group of 20s Mutually Influenced Countries ($M_{20}$), Fitness and the Real World Economic Blocks. On the left, (**a**) shows the $M_{20}$ network, code colored against the G7 (navy blue), extended G20 (all shades of blue), China (red) and countries outside the G20 economic block (olive). The middle figure (**b**) provides an indication of the intersection between the $M_{20}$ and countries with the highest fitness Φ (dark magenta). On the right, (**c**) shows the symbiotic nature of the edges within the $M_{20}$ network. Blue edges represent our definition of mutualism (where both countries reduce their fitness in the absence of the other), and pink links reflect competition (where both countries increase their fitness in the absence of the other), whereas cyan relates to reciprocal altruism (where one country reduces its fitness by the absence of the other, but the latter increases when the former is absent).

**Figure 5 entropy-25-00141-f005:**
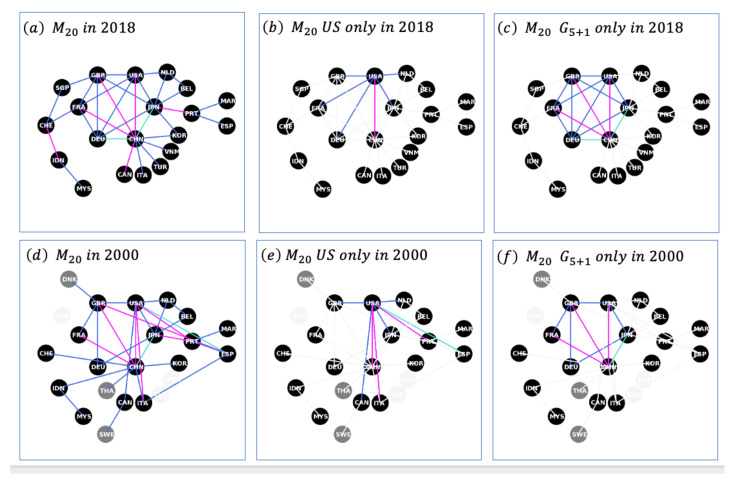
The Group of 20s Mutually Influenced Countries ($M_{20}$): 2018 vs. 2000. The left panels, (**a**–**d**), show the $M_{20}$ network in 2018 and 2000, respectively. Edges represent symbiotic relationships, where blue indicates mutualism, pink relates to competition, and cyan is associated with parasitism or reciprocal altruism. The grey nodes in panels (**d**–**f**) represent the countries that are no longer within the $M_{20}$ in 2018, whereas the fainting white—smoke nodes are related to the countries that were not part of the $M_{20}$ in 2000. The central panels, (**b**,**e**), consist of the edges solely related to the USA. Similarly, the right panels (**c**,**f**) represent edges solely associated with the founding G5 countries plus China.

## Data Availability

The analysis in this paper is based on data available here: https://efcdata.cref.it/integrated-database (accessed on 6 June 2022).

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
