# Peer review of "Economic Trading Susceptibility: Constructing Networks of Mutual Influence through the Fitness of Countries"

_entropy, 2023, doi:10.3390/e25010141_

Round 1
Reviewer 1 Report
General overview: The article is clearly written and well explained. I find this work very interesting and well motivated. The methods are solid and the results clear and well discussed but I have some comments that I would like to see addressed or explained before publication.
Important but philosophical:
The Fitness of a country is not absolute but rather a relative value compared to other countries. The whole point of the iterative algorithm is to try to measure the technological level of productive systems and the complexity of products by looking solely at the matrix Mcp, based on the idea that the hierarchy of the system is carrying that information within itself. I am sure the authors are well aware of this point, given the renormalization they use in eq.2 to compare Fitnesses. Nevertheless, a key point of the methodological foundation of this work is not discussed as it should be in my opinion. The influence here studied is based on how much the Fitness values change once we remove a country from the data. But what is the meaning of such change ? If we remove Germany from the database, are we saying that the capabilities of Spain or the complexity of shoes are actually changing or do we just lack important information to precisely estimate them from the data? In short, are the authors studying properties of the algorithm or of the economies? While I do consider this point important in the general discussion of this field, I do not think it undermines this work, but the authors should discuss it in the introduction.
Major:
eq.3 As the authors probably know, the values of F and Q are exponentially ‘distributed’. The linearity of eq.3 would require to use the logarithm of F in order to compare values so different, or equivalently the relative differences of the Fitness. To explain it better, if removing a country doubles a low-values Fitness (say F =10^-3), the absolute difference will still be much lower than a 5% increase of a high-values Fitness (F = 10). I think this point is very important and might affect their results greatly.
Minor
Eq.1 has a detail that differs from the original algorithm cited in [10]. Q is calculated using F at the nth step instead of (n-1)th. While results are the same for practical purpose and the form used in this work can actually be preferred (for reason I do not consider important here), the authors should point that difference out and briefly discus it. Regrettably, I am not aware of published papers to suggest for citation.
eq.2 As discussed, above a renormalization is needed in order to compare the fitness of two different matrices. In my opinion, this part is important for your results while being only briefly presented in the paper. I would ask the authors to expand the explanation of the equation and maybe to try one different renormalization to check the robustness of their results. For a binary Mcp one can add a dummy country with all 1s in its row, in order to set a constant reference during the time studied. Maybe one can find a similar solution for a weighted Mcp.
Fig.1 Usually, the lowest Fitnesses are converging to zero values. This is due to properties of the algorithm and its balancing. I can spot from Fig.1 that this might be the case for countries with log(Fitness) <-6,5 more or less. You may want to exclude these countries in your analysis. You can numerically identify them by looking at the vector S_c = sum_p (M_{cp}Q_p/F_c). By construction this vector should be constant, but non-converging countries show values different form the others.
text:
line-301 whit-> with
line 78 - “ it can be argued that products produced by a high-fitness country give limited information on the complexity of the product itself because these countries export almost all products.”
should read “it can be argued that high-fitness countries give limited information on the complexity of a product because these countries export almost all products.”
Reviewer 2 Report
Summary
In this paper, the authors focus on the study of the interconnection between the economies of different countries. They aim to quantify the mutual influence of countries using standard network analysis procedures. The main result is the development of a metric called Mutual Influence Robustness (MIR) which, when used in conjunction with the Economic Fitness metric, makes it possible to establish a predictive model of the evolution of different countries in the future. The results show a mutual influence between East and South East Asia countries, developed North America and Northern and Iberian countries.
Comments
The authors carry out a very interesting and ambitious study in which they seek to analyse the degree of mutual influence between countries through the network generated by the economic relations established between them. They begin their work with an analysis of the different approaches that have been used in the economic literature on this issue. They highlight the latest advances centred on the "fitness of a country" concept. The main novelty here is that their analysis considers additional methods and derived calculations based on changes in the metric rather than focusing on the values of the metric itself.
In chapter 2 the authors describe the algorithm used in their work. They consider the calculated values for the Fitness of the countries and using an iterative algorithm they calculate the nth iteration of the Fitness of the country and the complexity of the product. The key element in this iterative process is the matrix Mcp which represents the country-mix product that holds a value for the competitiveness of a country c in producing product p. This matrix is approximated by the market shares of the exported products. In this part, the authors use the formulas and data from the reference [14] cited in the text.
The most innovative part of the paper is analysed in sections 2.2 and 2.3, where the country influence matrix is created, based on the fitness of the country analysis, eliminating each country. To carry out the analysis, different threshold levels are established, eliminating the arcs that do not reach these levels; so that when the threshold increases, the symbiotic commensal relationships (where the influence of a country is small or neutral) tend to be eliminated first.
Finally, the authors develop the Mutual Influence Robustness (MIR) metric which considers that high values represent highly interweaving countries and small values of the MIR represent countries with commensalism relationships.
Once this index has been defined, the authors focus on an analysis of the relationships between the G20 countries.
Specific Comments
Some specifics questions to consider:
1. Page 3 – Equation (1) -> correct superscripts in the equation in the second bracket.
2. Page 5 – Line 215 -> What is the A in the expression of this line? The authors could improve the explanation of the relationship established in the paragraph.
3. Page 9 - Line 283 -> <space> between Figure and Number.
4. Page 10 - Line 311 -> <space> between Figure and Number.
Round 2
Reviewer 1 Report
I am satisfied with the changes on the paper and the authors’ arguments on my suggestions. One small thing: Figure 1 in the revised manuscript is still the old one, while the authors showed a new version in their rebuttal letter.
Happy new year